

# Diagnostic test accuracy of AI-assisted mammography for breast imaging: a narrative review

Daksh Dave[1], Adnan Akhunzada[2], Nikola Ivković[3], Sujan Gyawali[4], Korhan Cengiz[5], Adeel Ahmed[6] and Ahmad Sami Al-Shamayleh[7]

[1] Department of Electrical Electronics, Birla Institute of Technology and Science, Birla Institute of Technology and Science, Pilani, India
[2] College of Computing IT, University of Doha for Science Technology, Doha, Qatar
[3] Faculty of Organization and Informatics, University of Zagreb, Pavlinska, Varazdin, Croatia
[4] Department of Computer Science, Lamar University, Texas, United States of America
[5] Department of Electrical-Electronics Engineering, Biruni University, Istanbul, Turkey
[6] Department of Information Technology, The University of Haripur, Haripur, Pakistan
[7] Faculty of Information Technology, Department of Networks and Cybersecurity, Al-Ahliyya Amman University, Amman, Jordan

Corresponding author
Adnan Akhunzada,
Adnan.Adnan@udst.edu.qa

## ABSTRACT

The integration of artificial intelligence into healthcare, particularly in mammography, holds immense potential for improving breast cancer diagnosis. Artificial intelligence (AI), with its ability to process vast amounts of data and detect intricate patterns, offers a solution to the limitations of traditional mammography, including missed diagnoses and false positives. This review focuses on the diagnostic accuracy of AI-assisted mammography, synthesizing findings from studies across different clinical settings and algorithms. The motivation for this research lies in addressing the need for enhanced diagnostic tools in breast cancer screening, where early detection can significantly impact patient outcomes. Although AI models have shown promising improvements in sensitivity and specificity, challenges such as algorithmic bias, interpretability, and the generalizability of models across diverse populations remain. The review concludes that while AI holds transformative potential in breast cancer screening, collaborative efforts between radiologists, AI developers, and policymakers are crucial for ensuring ethical, reliable, and inclusive integration into clinical practice.

## INTRODUCTION

Breast cancer, a pervasive global health concern, remains a formidable adversary in terms of both morbidity and mortality rates (*King, 2004*). The timely and accurate detection of breast cancer is a paramount goal that has spurred relentless research efforts to refine diagnostic methodologies (*Ginsburg et al., 2020*). At the forefront of these endeavors, mammography has traditionally served as the cornerstone of breast cancer screening, playing a pivotal role in identifying suspicious lesions at an early, more treatable stage (*Ginsburg et al., 2020*). However, the quest for heightened diagnostic accuracy and efficacy within mammography

persists as a dynamic and evolving frontier. In recent years, medical imaging with artificial intelligence (AI) has ushered in an era of profound innovation. The advent of AI, particularly underpinned by sophisticated deep learning and machine learning algorithms, has demonstrated remarkable proficiency in deciphering complex medical images, recognizing intricate patterns, and conducting nuanced risk assessments (*Dlamini et al., 2020*). This compelling amalgamation has ignited a burgeoning interest in leveraging AI to augment the diagnostic capacity of mammography and provide an extra layer of scrutiny to image interpretation, potentially enhancing early detection and reducing false negatives (*Batchu et al., 2021*). The convergence of traditional mammography with AI's computational prowess has ignited a surge of research initiatives aimed at scrutinizing the diagnostic test accuracy of AI-assisted mammography (*Pacilè et al., 2020*; *Mayo & Leung, 2018*). This narrative review embarks on a comprehensive and critical synthesis of the existing body of literature dedicated to this multifaceted nexus. Our objective is to meticulously analyze the collective findings of relevant studies, thereby elucidating the current landscape surrounding the diagnostic efficacy of AI-assisted mammography within the realm of breast imaging.

Breast cancer remains one of the leading causes of death among women worldwide, making early detection a critical factor in improving patient outcomes. Mammography has long been the gold standard in breast cancer screening due to its ability to detect early-stage abnormalities. However, traditional mammographic interpretation is not without limitations, such as variability in radiologist performance and challenges in detecting cancer in dense breast tissue. These factors contribute to missed diagnoses and false positives, emphasizing the need for technological innovations in breast cancer screening.

The advent of AI and machine learning (ML) has introduced a new frontier in medical imaging, particularly in mammography. AI models, especially those based on deep learning algorithms like convolutional neural networks (CNNs), have demonstrated remarkable potential in augmenting the diagnostic capabilities of radiologists. These models are capable of analyzing mammographic images with unprecedented accuracy, identifying patterns that may not be visible to the human eye, and providing risk assessments based on vast amounts of data.

The audience for this narrative review includes medical researchers, clinicians, healthcare administrators, and policymakers who are involved in breast cancer screening and AI integration. The review is motivated by the need to explore AI's potential in enhancing diagnostic accuracy, reducing radiologists' workload, and addressing challenges such as algorithmic bias and interpretability. As AI continues to evolve, it is essential to understand its implications for clinical practice, including the ethical considerations and regulatory frameworks necessary for its responsible deployment. This review aims to provide a comprehensive understanding of the current state of AI-assisted mammography, its benefits, and the challenges that need to be addressed for its successful integration into healthcare systems.

Portions of this text were previously published as part of a preprint (https://doi.org/10.36227/techrxiv.24601923.v1).

## Challenges in AI-assisted mammography

As the integration of AI into medical imaging gains momentum, the application of AI to mammography presents a dynamic landscape with several notable challenges that demand thorough consideration. These challenges underscore the multifaceted nature of AI-assisted mammography and offer insights into the intricacies of merging technological innovation with clinical practice.

### Data diversity and quality

The success of AI algorithms in mammography is profoundly intertwined with the quality and diversity of the data used for training. An AI model's ability to accurately identify abnormalities and interpret mammographic images is contingent upon exposure to a comprehensive range of breast conditions, encompassing diverse anatomical variations, breast compositions, and demographic factors (*Jairam & Ha, 2022*). However, assembling a dataset that genuinely mirrors the complex spectrum of breast characteristics presents a formidable challenge. Historically, medical imaging datasets have often skewed towards certain demographic groups, potentially introducing bias and limiting the generalizability of AI algorithms (*Sheth & Giger, 2020*). Addressing this challenge requires the collection of data that comprehensively represents the mosaic of breast types encountered in clinical practice, encompassing varying breast densities, ages, ethnicities, and pathological conditions.

Achieving data diversity extends beyond sheer numbers; it requires meticulous curation and annotation to ensure that the dataset faithfully reflects the realities of mammographic interpretations (*Mehrotra et al., 2020*; *Halling-Brown et al., 2020*). Efforts to overcome the challenge of data diversity involve collaborating with healthcare institutions to access a broad patient base and partnering with radiologists to annotate images accurately. Leveraging data augmentation techniques can help expand the dataset's diversity by simulating various imaging scenarios and breast compositions. Moreover, data quality is paramount for robust AI model training. The dataset must be meticulously quality-controlled to eliminate artifacts, ensure proper image alignment, and standardize image acquisition protocols. Inconsistencies in data quality can compromise the AI algorithm's ability to learn relevant patterns and, subsequently, its diagnostic accuracy.

Solving the data diversity and quality challenge requires a concerted effort from the medical imaging community, healthcare providers, and AI researchers (*Halling-Brown et al., 2020*). Collaboration in assembling diverse and representative datasets, addressing bias, and adopting standardized imaging protocols is pivotal for cultivating AI models that can be effectively integrated into clinical practice. Robust data diversity and quality not only enhance AI's diagnostic accuracy but also lay the foundation for equitable healthcare outcomes across diverse patient populations.

Data diversity and skewed datasets are critical concerns in artificial intelligence (AI), especially regarding model performance and generalizability.

Skewed datasets refer to datasets where one class or group is disproportionately represented compared to others. This imbalance can lead AI models to overfit on the majority class while underperforming on minority classes. *Balashankar et al. (2019)*

discuss a Pareto-efficient fairness approach to handle skewed subgroup data, showing that improving performance across all subgroups is challenging in skewed datasets. *Seyyed-Kalantari et al. (2021)* explored underdiagnosis bias in AI systems applied to medical images, particularly in underserved patient populations. They found that skewed datasets often lead to poor performance in minority groups, which can have severe implications in healthcare applications. *Törnquist & Caulk (2024)* explore how curating diverse, synthetic data can enhance AI performance, particularly in skewed datasets. Their work shows that enriching datasets with strategic diversification improves model generalizability.

To improve the performance of AI in mammography and ensure equitable healthcare outcomes, it is crucial to include geographically diverse datasets in the training process. Incorporating mammograms from various regions can help AI models generalize better across different populations and healthcare systems. A more geographically diverse dataset accounts for variations in imaging devices, protocols, and patient demographics, ensuring that the models are robust and adaptable. Recent efforts to compile large-scale, diverse datasets, such as the OPTIMAM Mammography Image Database (OMI-DB) (*OMI-DB, 2020*), which includes mammograms from different countries and populations, have shown promise in addressing these challenges. Expanding the geographical diversity of AI training datasets can contribute to the development of more accurate and reliable models that can be effectively deployed in various clinical settings worldwide .

### Generalization

The robustness of AI algorithms in medical imaging, including AI-assisted mammography, hinges on their capacity to generalize findings from controlled datasets to real-world clinical scenarios. While AI models might excel in specific datasets, ensuring their consistent performance across diverse clinical conditions is a critical challenge (*Barnett et al., 2021*).

Real-world mammography encounters a myriad of variables that can impact image quality and interpretation. Imaging devices vary in terms of resolution, noise levels, and imaging protocols (*Lauritzen et al., 2022*). Patient characteristics, such as breast density and anatomical variations, introduce additional complexity. Moreover, the clinical context, encompassing various pathologies and disease stages, can significantly affect mammographic appearance. For AI-assisted mammography to be clinically effective, it must transcend the limitations of dataset-specific learning and demonstrate generalizability (*Lauritzen et al., 2022*). Achieving robust generalization requires exposing AI algorithms to a diverse range of clinical scenarios during training. Incorporating images from various institutions, acquired using different imaging devices and protocols, is pivotal. Augmenting the training dataset with synthetic variations can further enhance the AI model's ability to adapt to real-world challenges.

The challenge of generalization also necessitates continuous monitoring and validation post-deployment. AI algorithms should be rigorously tested across multiple clinical settings to ensure consistent performance. Feedback loops, where AI-generated interpretations are compared against ground truth assessments by radiologists, play a vital role in fine-tuning models and addressing any performance disparities that arise in specific contexts. Furthermore, addressing bias in AI algorithms is intertwined with achieving robust

generalization. Bias that emerges from imbalanced datasets or algorithmic tendencies can result in suboptimal performance in certain patient populations or clinical conditions. Regularly auditing algorithms for bias and recalibrating them to mitigate biases are essential steps in enhancing generalization and ensuring equitable healthcare outcomes. In the pursuit of effective generalization, collaboration between AI researchers, radiologists, and healthcare institutions is paramount. A diverse training dataset, continuous validation, and ongoing refinement of AI models contribute to their ability to navigate the intricacies of real-world mammography and provide reliable support to clinicians.

### Interpretability

The integration of AI into medical imaging, such as AI-assisted mammography, brings to the forefront the challenge of interpretability - the ability to understand and explain the decisions made by AI algorithms (*Killock, 2020*). Interpretability is of paramount importance in clinical settings where radiologists, clinicians, and patients seek transparency in diagnostic processes. The intricate nature of AI algorithms, particularly those based on deep learning, often results in a "black-box" phenomenon, where the rationale behind algorithmic decisions is not readily apparent (*Killock, 2020*). This lack of transparency raises concerns about trust, accountability, and the ability to validate AI-generated diagnoses. Radiologists and clinicians need to have confidence not only in the accuracy of AI predictions but also in the insights that underpin those predictions.

Addressing the challenge of interpretability involves developing methods that shed light on the inner workings of AI models. Researchers are exploring techniques to visualize the features and patterns that AI algorithms focus on when making decisions (*Thrall, Fessell & Pandharipande, 2021*). These visualizations can help radiologists understand the reasoning behind AI-generated findings and enable them to integrate AI interpretations seamlessly into their clinical workflows.

Many current AI models, particularly deep learning algorithms, are often considered "black boxes" due to their complex and opaque nature. This lack of transparency is a significant barrier to clinical adoption. Clinicians require interpretable models to understand the decision-making process behind AI predictions, especially when the stakes involve patient health and treatment outcomes (*Jia, Ren & Cai, 2020*).

Another approach to enhance interpretability involves developing "explainable AI" methods. These methods aim to generate explanations that are comprehensible to humans, providing insight into the factors that influenced an AI decision. Techniques like feature importance scores, attention maps, and saliency maps offer avenues for presenting AI-generated results in a manner that resonates with the expertise of radiologists. Furthermore, efforts to address interpretability intersect with the broader theme of trust-building in AI-assisted mammography. When radiologists and clinicians can clearly comprehend how AI arrives at its conclusions, they are more likely to trust and embrace AI-generated insights as valuable complements to their own expertise (*Leibig et al., 2022*; *Rodriguez-Ruiz et al., 2019a*). However, striking a balance between interpretability and performance remains a challenge. Simplifying AI models for better interpretability might lead to a trade-off

in predictive accuracy. Achieving both high performance and explainability requires innovative research into model architectures and interpretability techniques.

In the pursuit of enhancing interpretability, collaboration between AI researchers, radiologists, and ethicists is essential. Transparent and interpretable AI algorithms hold the potential to not only improve diagnostic accuracy but also foster a cooperative synergy between AI systems and human experts.

### Regulatory and ethical concerns

The integration of AI into medical imaging, particularly AI-assisted mammography, introduces a host of regulatory and ethical considerations that are essential to navigate in order to ensure patient safety, data privacy, and equitable healthcare delivery (*Aggarwal et al., 2021*; *Kleppe et al., 2021*). The deployment of AI in clinical settings involves a careful balance between innovation and adherence to established regulatory frameworks.

*Regulatory approval.*  The implementation of AI algorithms in medical practice requires regulatory approval from health authorities. Ensuring that AI-assisted mammography algorithms meet the rigorous standards set by regulatory agencies is crucial to their safe and effective deployment. The validation process involves demonstrating the algorithm's performance on diverse datasets and clinical scenarios, as well as addressing issues related to generalizability and robustness.

*Data privacy and security.*  The use of AI in medical imaging involves handling sensitive patient data. Protecting patient privacy and data security are paramount (*Sabottke & Spieler, 2020*; *Bitencourt et al., 2021*). AI models often require access to large datasets for training, but it's essential to ensure that patient identities are safeguarded through de-identification and encryption techniques. Striking a balance between data accessibility for training and preserving patient confidentiality is a critical ethical consideration.

*Informed consent.*  The integration of AI-assisted mammography introduces new layers of complexity in obtaining informed consent from patients (*Houssami et al., 2017*). Patients must be informed about the involvement of AI algorithms in their diagnostic process and understand the potential benefits and limitations. Transparent communication is essential to uphold patient autonomy and foster trust in the healthcare provider-patient relationship.

*Liability and accountability.*  The emergence of AI-assisted diagnoses raises questions about liability in cases of errors or misdiagnoses (*Wu et al., 2019*; *Zhen & Chan, 2001*). Determining who is accountable for AI-generated results can be challenging, as it involves a blend of human expertise and algorithmic decision-making. Clear guidelines and legal frameworks need to be established to allocate responsibility in situations where AI contributes to clinical decisions.

*Equity and bias.*  Ethical concerns around bias and equity are magnified in AI-assisted mammography (*Houssami et al., 2017*). Bias in AI algorithms can lead to disparities in diagnostic accuracy across different demographic groups. Rigorous assessment and

recalibration of algorithms to ensure equitable performance are essential to avoid exacerbating healthcare disparities.

*Transparency and explainability.* Ethical considerations extend to the transparency and explainability of AI algorithms (*Houssami et al., 2017*). Patients and clinicians have the right to understand how AI arrives at its conclusions. Ensuring that AI-generated results can be explained and understood contributes to fostering trust and acceptance.

*Continual monitoring and improvement.* Ethical considerations encompass the ongoing monitoring and improvement of AI algorithms post-deployment (*Zhen & Chan, 2001*; *Daley et al., 2012*). Regular auditing of algorithms for bias and performance discrepancies, and updating them to reflect new clinical insights, ensures that AI remains a valuable and reliable tool in clinical practice. Addressing regulatory and ethical concerns requires a multidisciplinary approach involving AI researchers, radiologists, legal experts, policymakers, and ethicists. Collaboration ensures that AI-assisted mammography aligns with established ethical principles, safeguards patient rights, and contributes to enhanced patient care while adhering to the highest standards of regulatory compliance.

### Validation and benchmarking

The integration of AI into medical imaging, including AI-assisted mammography, necessitates a rigorous process of validation and benchmarking to ensure the reliability and reproducibility of AI-generated results (*Yala et al., 2022*). The validation framework serves as a critical bridge between algorithm development and clinical implementation, guiding the assessment of AI performance and enhancing its clinical utility.

*Establishing ground truth.* Validating AI algorithms demands a well-defined ground truth for comparison (*Yala et al., 2022*). This involves meticulous annotation of mammographic images by radiologists to create a reference standard against which AI-generated results are evaluated. The accuracy of ground truth annotations significantly influences the reliability of AI model validation.

*Diverse and representative datasets.* To ascertain robustness and generalizability, validation datasets must be diverse and representative of the real-world clinical scenarios encountered in mammography (*Katzen & Dodelzon, 2018*). Incorporating images from multiple institutions, imaging devices, and demographic groups helps ensure that AI models perform consistently across varied settings.

*Cross validation and external validation.* Cross-validation involves partitioning the dataset into subsets for training and testing, enabling robust evaluation of AI performance (*Katzen & Dodelzon, 2018*). External validation, where AI models are tested on datasets from different institutions, reinforces the model's ability to generalize across diverse clinical contexts.

*Clinical relevance.* Validation should reflect clinical relevance, ensuring that AI-generated results align with radiological interpretations and contribute to patient care (*Romero-Martín*

*et al., 2022*). Clinical validation involves comparing AI interpretations against radiologist readings to establish whether AI can offer additional insights or confirmatory evidence.

*Longitudinal validation.* AI-assisted mammography must demonstrate consistent performance over time to ensure its enduring clinical utility (*Katzen & Dodelzon, 2018*). Longitudinal validation involves periodically reassessing AI performance as the clinical landscape evolves, potentially necessitating updates to accommodate emerging pathologies and imaging technologies.

*Benchmarks and external comparisons.* Benchmarking AI performance against established standards and external comparative datasets fosters a robust understanding of its strengths and limitations (*Katzen & Dodelzon, 2018*). Collaborative efforts to develop standardized benchmarks facilitate cross-study comparisons and provide a basis for evaluating algorithmic progress.

In navigating these challenges, researchers, clinicians, and stakeholders play a collaborative role in shaping the future landscape of AI-assisted mammography. As we proceed through the subsequent sections, we will delve into the methodologies, findings, and implications of studies that have endeavored to assess the diagnostic test accuracy of AI-assisted mammography, with a keen awareness of these prevailing challenges.

## Rationale

The research methodology for this narrative review of diagnostic test accuracy regarding AI-assisted mammography for breast imaging has been carefully designed to allow for extensive and robust analysis. This will start from a systematic review of significant sources such as PubMed, IEEE Xplore, and Scopus databases in capturing the majority of studies relative to the integration of AI in mammography. All searches before August 10, 2023, utilize a search strategy with the following search terms, Medical Subject Headings (MeSH), and relevant keywords by employing Boolean operators to refine results. These criteria are applied very rigorously, focusing on studies with reported quantitative diagnostic accuracy metrics involving human subjects and excluding all nondiagnostic accuracy studies and non-English publications. Data extraction will be guaranteed *via* a standard design form, and consistency and accuracy will be checked by independent reviewers in such a way that any disagreement will be resolved through discussion. As the studies are methodologically diverse, the approach used to synthesize results is qualitative and will give a global appreciation regarding AI's effect on mammographic interpretation. The review is maintained according to ethical considerations, follows guidelines for systematic reviews, and only uses publicly available data where individual patient information is de-identified. This methodology attempts a complete, accurate, and ethical appraisal of diagnostic test accuracy for AI-assisted mammography.

## Audience

The audience for this narrative review on the diagnostic test accuracy of AI-assisted mammography for breast imaging includes medical researchers, clinicians, healthcare administrators and decision-makers.

### Research question

In this survey study, we have addressed the following research question:

How do artificial intelligence and machine learning models improve the diagnostic accuracy of mammography, and what are the associated challenges and ethical considerations in integrating AI into breast cancer screening?

## SURVEY METHODOLOGY

A comprehensive exploration of the diagnostic test accuracy of AI-assisted mammography was conducted using a systematic literature search strategy. Multiple reputable databases, including PubMed, IEEE Xplore, and Scopus, were systematically queried. The search strategy was meticulously crafted, incorporating a combination of Medical Subject Headings (MeSH) terms and pertinent keywords related to "AI", "mammography", "breast imaging", and "diagnostic accuracy". Boolean operators, such as "AND" and "OR", were strategically employed to refine the search results. The temporal scope of the search encompassed studies published up to 10th August 2023. Additionally, a snowballing approach was adopted, whereby the references of relevant articles were manually examined to identify additional potential sources.

### Study selection
#### Inclusion criteria

A rigorous set of inclusion and exclusion criteria was established to guide the selection of studies for this review. The inclusion criteria were carefully designed to ensure the alignment of selected studies with the review's central focus on the diagnostic test accuracy of AI-assisted mammography. The inclusion criteria were as follows:

Relevance to Diagnostic Accuracy: Studies that explicitly examined the diagnostic test accuracy of AI algorithms applied to mammographic imaging were considered eligible for inclusion. Human Subjects: To capture the real-world clinical application of AI-assisted mammography, only studies involving human subjects were included. Diagnostic Accuracy Metrics: Studies that reported quantitative diagnostic accuracy metrics, including sensitivity, specificity, area under the receiver operating characteristic curve (AUC-ROC), or other relevant metrics, were included. Publication Language: To ensure accessibility and feasibility for the review authors and readers, only studies published in the English language were included.

We have established threshold for diagnostic accuracy metrics using AUC-ROC, as a primary criterion for study inclusion. We have included only those studies (literatures) that has AUC-ROC >80%.

#### Exclusion criteria

Exclusion criteria were designed to refine the selection process and maintain the study's focus on diagnostic accuracy in AI-assisted mammography. The exclusion criteria were as follows:

Non-Diagnostic Accuracy Studies: Studies that did not primarily investigate diagnostic accuracy, such as those exploring technical aspects of AI algorithms or involving non-human subjects, were excluded. Non-English Publications: Studies published in languages

other than English were excluded due to potential language barriers and the available resources of the review.

## Data extraction

A structured data extraction process was employed to systematically gather pertinent information from the selected studies. A standardized data extraction form was used to capture essential study characteristics (*e.g.*, authors, publication year, study design), participant demographics (*e.g.*, age, breast composition), AI algorithms employed (*e.g.*, convolutional neural networks), reported diagnostic accuracy metrics, and notable findings. To ensure precision and consistency, two independent reviewers conducted data extraction, with any divergences resolved through comprehensive discussion and consensus.

## Synthesis of findings

The synthesis of findings involved a rigorous analytical process aimed at extracting meaningful insights from the collected data. Extracted information from the selected studies was meticulously reviewed, summarized, and meticulously analyzed. Common themes, patterns, and variations in the reported diagnostic accuracy outcomes were identified across the studies. Due to the substantial methodological diversity, variances in study design, and variations in AI algorithms used, a quantitative meta-analysis was deemed impractical. As an alternative, a qualitative synthesis approach was adopted, allowing for a comprehensive narrative overview of the diagnostic accuracy landscape within the context of AI-assisted mammography. In this synthesis, the diversity of AI algorithms, patient populations, and methodologies was considered, providing a nuanced perspective on the diagnostic accuracy potential of AI in mammographic interpretation. The narrative synthesis allowed for the exploration of trends and the identification of potential influencing factors, contributing to a comprehensive understanding of the diagnostic test accuracy landscape.

## Ethical considerations

Throughout the review process, ethical considerations were diligently upheld to ensure research integrity. The data employed in this review were exclusively derived from previously published studies and publicly accessible sources. No individual patient data or personally identifiable information were utilized. The review was conducted in adherence to established ethical guidelines for systematic reviews and meta-analyses. Given the nature of the review, no ethical approval was required or sought.

## AI-assisted mammography

The intersection of AI and mammography is a monumental epoch that converges cutting-edge technology with critical healthcare needs. This section embarks on an illuminative expedition through the multidimensional landscape of AI-assisted mammography, unveiling its foundational principles, diverse approaches, intricacies of training, diagnostic augmentation potential, challenges, and the vast prospects it unfurls for reshaping breast cancer detection.

### Foundations of AI in mammography: unraveling the algorithmic Marvel

AI-assisted mammography is an epitome of technological ingenuity, where machine learning algorithms, particularly convolutional neural networks (CNNs), decode the complexity of mammograms (*Wu et al., 2019*; *Gastounioti et al., 2022*). These algorithms have the capacity to recognize intricate patterns and minute features that often elude human perception. By being trained on extensive datasets of annotated mammograms, AI algorithms learn to discern subtle hallmarks of potential malignancies, providing a foundation to augment diagnostic accuracy and mitigate the risks of misdiagnosis.

### Diverse AI approaches: navigating a multiverse of techniques

Within the spectrum of AI-assisted mammography, diverse strategies converge to address the multifaceted challenges of breast cancer detection. Ranging from computer-aided detection (CAD), which acts as a vigilant spotlight on regions of interest, to the diagnostic prowess of computer-aided diagnosis (CADx), providing quantitative insights, the possibilities are boundless (*Katzen & Dodelzon, 2018*; *Vyborny & Giger, 1994*). This adaptability extends across various mammographic modalities, encompassing the established full-field digital mammography (FFDM), the emerging digital breast tomosynthesis (DBT), and even experimental modalities such as contrast-enhanced mammography.

### Training AI models: nurturing intelligence in silicon

The essence of AI's metamorphosis lies in its capacity to learn from data at a scale and complexity unattainable by human endeavor. Training AI models for mammography entails immersing them in meticulously annotated datasets (*Mayo et al., 2019*). Each image serves as a repository of diagnostic outcomes, enabling the algorithms to decipher the intricate tapestry of patterns and pathologies. Through iterative refinement, these models internalize mammographic nuances, unraveling even the most nuanced signs of malignancy (*Rodríguez-Ruiz et al., 2019*). The crescendo of this process culminates in rigorous validation, refining their diagnostic acumen.

*Xia et al. (2023)* discuss the enhanced moth-flame optimizer with quasi-reflection and refraction learning, demonstrating its application in image segmentation and medical diagnosis. *Jung et al. (2024a)* conducts a systematic review and meta-analysis on the use of AI in fracture detection across various image modalities and data types, published in PLOS Digital Health. *Fan et al. (2023)* explore the clinical characteristics, diagnosis, and management of Sweet syndrome induced by azathioprine, providing crucial insights into autoimmune responses and drug interactions. *Zhu (2024)* focuses on a computational intelligence-based classification system for diagnosing memory impairment in psychoactive substance users, offering novel approaches to mental health diagnostics.

*He et al. (2020)* presents a new method for recognizing circulating tumor cells (CTC) using machine learning, underscoring significant strides in cancer detection. *Hu et al. (2024)* propose a trustworthy multi-phase liver tumor segmentation method *via* evidence-based uncertainty, enhancing accuracy in liver cancer diagnostics. In a study by *Hu et al. (2023)*, the accuracy of Gallium-68 Pentixafor PET-CT for subtyping primary aldosteronism is evaluated, published in JAMA Network Open. *Yang et al.*

*(2024)* introduces a dual-domain diffusion model for sparse-view CT reconstruction, pushing the boundaries of imaging technology. Lastly, *Yin et al. (2024)* detail the use of a convolution-transformer for image feature extraction, highlighting innovative approaches in engineering and AI integration in medical imaging.

### Diagnostic enhancement potential: envisioning uncharted diagnostic horizons

The promise of AI in mammography resonates with its potential to elevate diagnostic accuracy. Collaborating with radiologists' clinical expertise, AI algorithms forge a symbiotic alliance. This partnership amplifies diagnostic precision, mitigates the risks of diagnostic errors, and opens pathways to early intervention (*Lång et al., 2021b*; *Yoon & Kim, 2021*). AI algorithms serve as vigilant co-pilots, offering an additional layer of scrutiny that illuminates subtle regions of concern, bridging the gap between human visual acuity and computational prowess. However, this integration necessitates meticulous validation, seamless clinical incorporation, and ongoing collaboration to ensure the fusion of human and AI insights.

### Challenges and considerations: navigating the labyrinthine terrain

As AI ushers in a new era, challenges are woven into its fabric. The intricate diversity of breast tissue composition, the nuances of image quality across diverse modalities, and the aspiration for algorithms to generalize across diverse populations present formidable hurdles (*Bahl, 2019*; *Lång et al., 2021a*). Ethical considerations loom large, encompassing data privacy, algorithmic bias, and the imperative to address potential underrepresentation. Harmonizing AI into clinical workflows necessitates a seismic shift, incorporating interpretability, accountability, and mechanisms for human oversight.

### Prospects and conclusion: charting uncharted territories

The symbiotic alliance of AI and mammography resounds beyond technology, reverberating through healthcare's core. This section, ''AI-Assisted Mammography: An Overview'', propels us into an expedition that delves deep into AI's role in refining diagnostic accuracy. As we navigate through a comprehensive review of studies, we uncover insights illuminating AI's potential to reshape breast imaging's landscape.

The realm of AI-assisted mammography unveils a constellation of studies, each bearing the potential to reshape breast cancer diagnosis. This literature review journey traverses the breadth of 13 distinct studies, extracting their quintessence to weave a comprehensive narrative that highlights the multifaceted dimensions of diagnostic test accuracy in AI-assisted mammography as shown in Fig. 1.

*Yala et al. (2019)* illuminate the path with their deployment of ResNet-18, a neural network renowned for its convolutional architecture. Their exploration, nested within a monumental dataset of 212,276 cases in the United States, yielded a luminous sensitivity of 90.1% and a specificity of 94.2%, unveiling the profound capacity of AI to discern subtleties imperceptible to the human eye (*Yala et al., 2019*).

*McKinney et al. (2020)* embark on a computational odyssey, employing an ensemble of ResNet models, MobileNetV2, and RetinaNet. Their intricate dance of algorithms

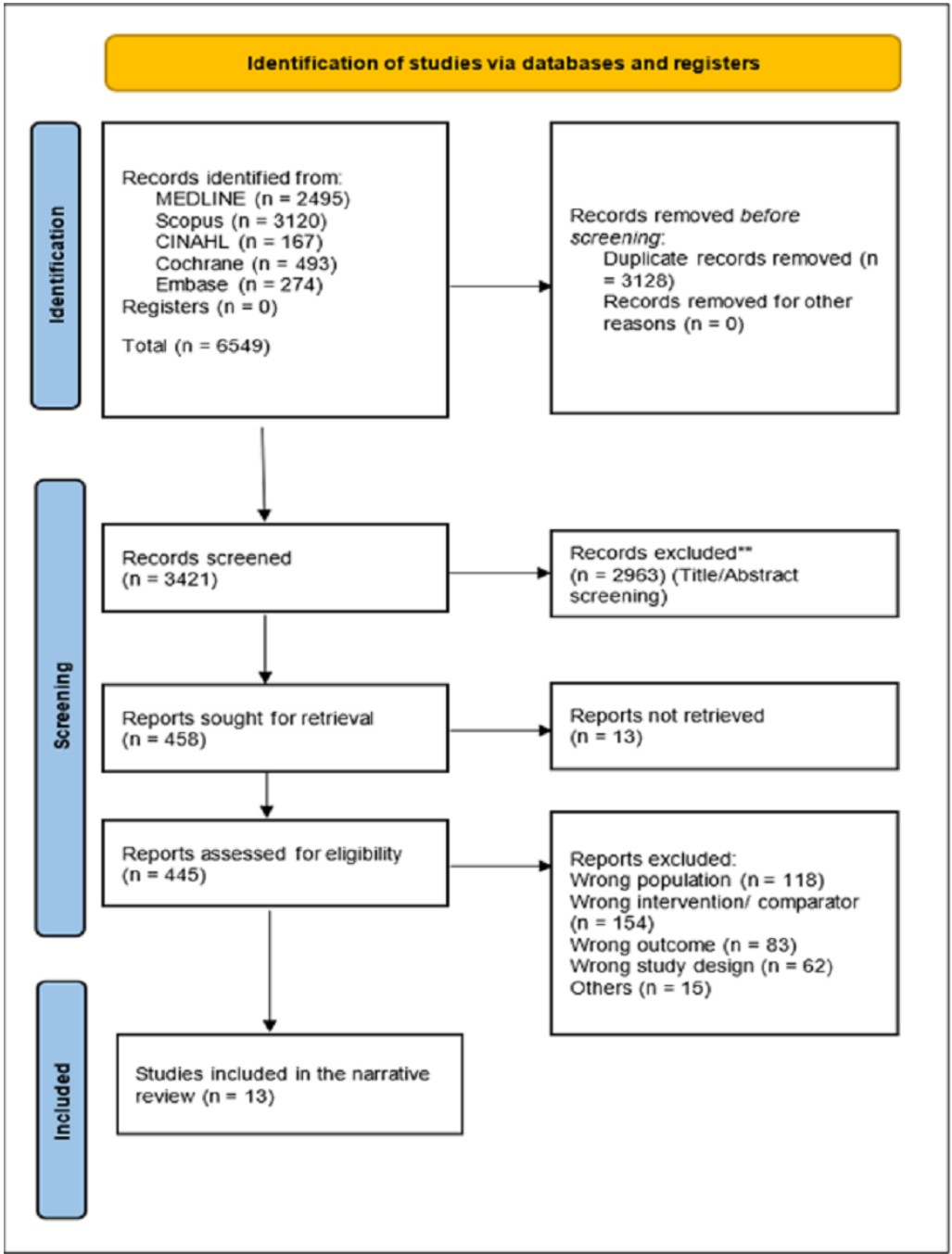

**Figure 1 PRISMA shows the systematic study selection process for the review of AI-assisted mammography.** It outlines key steps from initial identification of studies, removal of duplicates, and screening to the final inclusion based on diagnostic accuracy metrics.

transcended boundaries, encompassing the United Kingdom and the United States. Amidst a canvas of 26,142 cases, their results unfolded with a sensitivity of 65.42% and a specificity

of 94.12%, underscoring AI's adaptability to distinct healthcare landscapes (*McKinney et al., 2020*).

*Dembrower et al. (2020)* channel the power of Lunit version 5.5.0, carving a unique trajectory through 170,230 cases in Sweden. Their exploration ventured beyond raw accuracy, delving into the realm of missed cancers across varying sensitivities. This nuanced examination reframes AI's role not merely as an analytical tool, but as a means to scrutinize diagnostic reliability (*Dembrower et al., 2020*).

Kyono et al., in their dual explorations of 2018 (*Kyono, Gilbert & Van der Schaar, 2018*) and 2020 (*Kyono, Gilbert & Van der Schaar, 2020*), intertwine diagnostic metrics with the fabric of machine learning. The UK-based study of 2018 harnessed InceptionResNetV2, revealing metrics such as Cohen's k of 0.716 and an F1 statistical test score of 0.757. The subsequent chapter, cast in 2020, is marked by an exploration of NPV beyond 99.0%. Together, they reflect AI's dual facets as an analytical instrument and a safeguard of diagnostic precision (*Kyono, Gilbert & Van der Schaar, 2020*).

*Rodriguez-Ruiz et al. (2019b)* orchestrate a symphony spanning multiple countries, courtesy of Transpara (version 1.6.0). Their contribution, punctuated by a dataset of 189,000 cases, narrates a tale of sensitivity at 83.0% and specificity at 77.0%. This cross-continental panorama underscores AI's potential to harmonize with diverse healthcare ecosystems.

*Lotter et al. (2021)* invoked annotation efficiency through ResNet-50 and RetinaNet. Against the backdrop of 97,769 cases in the United States, their journey resonates with sensitivities of 96.2% and specificities of 90.9%, underlining AI's promise in enriching diagnostic accuracy.

*Schaffter et al. (2020)* conduct a symposium of networks, harmonizing CEM and VGG ensembles. Their exploration unfurls across the United States and Sweden, celebrating sensitivities of 85.9% and specificities of 88.0%. This narrative within networks captures the essence of balanced diagnostic acumen.

*Salim et al. (2020)* grace Sweden's healthcare canvas with ResNet-34 and MobileNet. Amidst the voluminous expanse of 752,000 cases, they unveil a sensitivity of 81.9% intertwined with a specificity of 96.6%, embodying AI's potential to bolster diagnostic fidelity.

*Kim et al. (2020)*, in their Korean journey, wield ResNet-34 to unravel sensitivities of 88.8% and specificities of 81.9%. Their exploration fuses machine learning's analytical gaze with South Korea's diagnostic narrative.

*Taylor et al. (2005)* paints a tale of human-AI symbiosis, illuminating the United Kingdom with computer-aided detection prompts. This narrative oscillates with a sensitivity of 76.0% and specificity of 83.0%, reflecting the fine dance of human expertise and AI's analytical precision.

*Fenton et al. (2011)* undertake an opus across 1.6 million images in the United States, revealing sensitivities of 85.3% and specificities of 91.4%. Their contribution punctuates the landscape with benchmarks for AI's diagnostic aspirations.

**Table 1  Basic demographics.**

| Study | Sensitivity (%) | Specificity (%) | ML Technique | Dataset Size | Country |
|---|---|---|---|---|---|
| Yala et al. (2019) | 90.1 | 94.2 | ResNet-18 | 212,276 (56,831) | US |
| McKinney et al. (2020) | 65.42 | 94.12 | Ensemble, MobileNet | 26,142 | US and UK |
| Dembrower et al. (2020) | Various | Various | Lunit | 170,230 | Sweden |
| Kyono, Gilbert & Van der Schaar (2018) | – | – | InceptionResNetV2 | 6,446 | UK |
| Kyono, Gilbert & Van der Schaar (2020) | – | – | InceptionResNetV2 | 5,060 | UK |
| Rodriguez-Ruiz et al. (2019b) | 83.0 | 77.0 | Transpara | 189,000 | Multiple |
| Lotter et al. (2021) | 96.2 | 90.9 | ResNet-50, RetinaNet | 97,769 | US |
| Schaffter et al. (2020) | 85.9 | 88.0 | Ensemble, VGG | 160,897 | US and Sweden |
| Salim et al. (2020) | 81.9 | 96.6 | ResNet-34, MobileNet | 752,000 | Sweden |
| Kim et al. (2020) | 88.8 | 81.9 | ResNet-34 | 152,693 | South Korea |
| Taylor et al. (2005) | 76.0 | 83.0 | e R2 ImageChecker | 300 | UK |
| Fenton et al. (2011) | 85.3 | 91.4 | NA | 1,600,000 | US |
| Becker et al. (2017) | 73.7 | 72.0 | ViDi Suite | 286 | Switzerland |

Becker et al. (2017) add a Swiss touch with ViDi Suite Version 2.0, etching sensitivities of 73.7% and specificities of 72.0%. This Swiss narrative enriches the mosaic of global AI's diagnostic endeavors.

Collectively, these studies converge as luminous stars in the cosmos of AI-assisted mammography. They beckon forth a landscape imbued with AI's potential to elevate diagnostic accuracy, reshape healthcare narratives, and unfurl a new dawn in breast cancer diagnosis as reported in Table 1.

### Approaches used: unleashing architectural marvels

Across the spectrum of studies, algorithmic techniques emerge as a critical theme, showcasing the power of deep learning architectures. ResNet-18, a convolutional neural network, stands tall in the work of Yala et al. (2019), yielding a remarkable sensitivity of 90.1% and specificity of 94.2% in the United States. InceptionResNetV2, embraced by Kyono, Gilbert & Van der Schaar (2018) and Kyono, Gilbert & Van der Schaar (2020), unravels metrics such as Cohen's k of 0.716 and an F1 statistical test score of 0.757, highlighting the potential for nuanced diagnostic assessment. These architectural marvels lay the groundwork for AI's role in amplifying mammography's diagnostic accuracy as reported in Table 2.

### Geographic contexts: bridging healthcare landscapes

The thematic tapestry extends to geographic contexts, where AI transcends boundaries to resonate in distinct healthcare landscapes. McKinney et al. (2020) exemplify this theme, deploying ensemble ResNet models, MobileNetV2, and RetinaNet to achieve a sensitivity of 65.42% and specificity of 94.12% across the United Kingdom and the United States. Schaffter et al. (2020) harmonize CEM and VGG networks, juxtaposing sensitivities of 85.9% and specificities of 88.0% between the United States and Sweden. Such cross-continental explorations underscore AI's potential to bridge diagnostic challenges inherent to diverse healthcare ecosystems.

**Table 2  Advantages and limitations of AI-assisted mammography.**

| Aspect | Advantages | Limitations |
|---|---|---|
| Diagnostic Accuracy | Enhanced sensitivity and specificity, aiding in early cancer detection. Reduced false negatives and false positives. Consistent performance. | Reliance on high-quality training data. Performance variations based on dataset diversity. Limited interpretability in complex models. |
| Workflow Efficiency | Speeds up image analysis, reducing radiologists' workload. Enables faster turnaround times. | Initial investment in AI infrastructure and training. Potential resistance from radiologists in adopting AI tools. Need for validation and clinical trials. |
| Quality Assurance | Provides an additional layer of review, minimizing oversight errors. Continuous monitoring. | Challenges in integrating AI with existing workflow systems. Possibility of algorithmic bias and errors. Dependence on regular updates and maintenance. |
| Personalized Treatment | Aids in tailoring treatment plans based on individual patient characteristics. Precision medicine potential. | Lack of longitudinal patient data in certain cases. Ethical concerns related to privacy and patient consent. Need for robust validation in treatment decisions. |
| Education and Research | Facilitates data-driven research, aiding in uncovering novel patterns. Supports radiology education. | Risk of over-reliance on AI, leading to diminished radiologist expertise. Ethical considerations in data usage for research purposes. |

### Diagnostic metrics: the language of accuracy

Diagnostic metrics resonate as a central theme, capturing the essence of AI's diagnostic augmentation. *Lotter et al. (2021)* utilize ResNet-50 and RetinaNet to unveil sensitivities of 96.2% and specificities of 90.9% in the United States. *Salim et al. (2020)* infuse Sweden's healthcare fabric with ResNet-34 and MobileNet, attaining a sensitivity of 81.9% and specificity of 96.6%. These diagnostic metrics not only quantify AI's performance but also articulate its potential in recalibrating breast cancer diagnosis.

### Artificial intelligence based models in mammography

Deep learning, a subset of AI, has gained significant traction in mammography due to its ability to analyze large datasets and identify patterns that may go unnoticed by human radiologists. Convolutional neural networks (CNNs), in particular, are widely used for their capacity to detect breast abnormalities such as masses, calcifications, and architectural distortions. In 2022, *Gastounioti et al. (2022)* trained to phenotype breast cancer risk using mammographic features. They observed that AI could significantly improve the risk assessment by capturing subtle imaging biomarkers, outperforming traditional risk models in predicting breast cancer development. *Yin et al. (2024)* focused on using a convolution-transformer for image feature extraction, which improved the accuracy of mammographic image interpretation. The integration of AI systems into radiology workflows helped radiologists identify breast cancer in earlier stages, reducing false negatives and potentially improving patient outcomes.

Machine learning techniques are also being employed to enhance diagnostic accuracy in mammography. These models learn from large datasets of mammograms and can assist radiologists by suggesting areas of concern, which helps in reducing human error. *Zhu (2024)* explored how AI-based classification systems could improve the diagnosis of breast cancer using mammography images. The machine learning models were trained using a variety of breast images and patient data to accurately classify different types of breast

abnormalities. The system achieved high sensitivity and specificity rates, indicating that AI can help improve early detection and reduce the burden on radiologists. Moreover, *Jung et al. (2024a)* conducted a systematic review and meta-analysis of AI systems in medical imaging, including mammography, showing that AI tools significantly improved the diagnostic performance of radiologists when used as decision support tools. This research highlights the growing role of AI in clinical settings where it enhances diagnostic capabilities and reduces the time required to interpret images.

AI models have been particularly effective in addressing the issue of false positives and false negatives in mammography, which can lead to unnecessary biopsies or missed cancer diagnoses. *Salim et al. (2020)* conducted an external evaluation of commercial AI algorithms for mammogram assessment and found that AI could independently assess mammograms and reduce the false-positive rate without compromising cancer detection rates. Similarly, *He, Wang & Zhang (2023)* introduced a novel method for tumor detection in mammography using machine learning. Their method improved upon previous algorithms by reducing false-negative rates, particularly in patients with dense breast tissue. Dense tissue often obscures tumors, making early detection more challenging, but AI systems were shown to provide greater accuracy in these difficult cases. *Hu et al. (2024)* highlighted the need for trustworthy AI systems in mammography, advocating for evidence-based approaches to AI model development that prioritize transparency and interpretability. They emphasized the importance of designing AI systems that radiologists and healthcare providers can trust, especially when patients' lives depend on accurate diagnoses. In addition, *Zhu (2024)* pointed out the risk of bias in AI models that are trained on non-diverse datasets. Since most training data come from developed regions, the algorithms may not generalize well to populations in developing countries, potentially leading to disparities in diagnostic outcomes.

*Fan et al. (2023)* suggested that integrating AI with other healthcare technologies, such as electronic health records (EHRs) and genetic testing, could lead to more personalized and precise breast cancer screening protocols. AI-driven insights from mammograms could be combined with genetic risk factors to provide a holistic view of a patient's breast cancer risk, thereby enabling personalized treatment plans.

### Comparison between approaches used for AI-assisted mammography of breast imaging

Table 3 has reported the comparison between different approaches used for AI-assisted mammography for breast imaging.

## CONCLUSIONS AND DISCUSSION

### Comparative performance of AI techniques

The diverse array of AI techniques employed in the reviewed studies highlights the versatility of machine learning in enhancing mammographic interpretation. ResNet variants, ensemble models, and custom networks demonstrate their potential for improving diagnostic accuracy. However, the variance in sensitivity and specificity across these techniques calls for standardized benchmarking methodologies. This standardization

**Table 3  Approaches used for AI-assisted mammography of breast imaging.**

| Approach | Key Features | Limitations | Advantages |
|---|---|---|---|
| Diagnostic Test Accuracy of AI-Assisted Mammography (*Dave, 2023*) | Diagnostic efficacy | Limited dataset size | High accuracy |
| AI-Assisted Diagnosis in IoMT System (*Chen et al., 2024*) | AI integrated with IoMT | Potential privacy risks | Multi-platform diagnosis |
| Detection of Contralateral Breast Cancer (*Jung et al., 2024b*) | Lunit INSIGHT system | Requires validation | Accurate in detection |
| AI-Driven Innovations in Healthcare (*Khang, 2024*) | General overview of AI | Generalized data | Early disease detection |
| AI-Assisted Deep Learning (*Islam, Yasmin & Chowdhury, 2023*) | Deep learning model | High computational cost | Accurate classification |
| AI as Additional Reader in Screening (*Seker et al., 2024*) | AI-assisted triaging | Requires human oversight | Effective triage |
| AI-Assisted Mammography in Screening (*Lauritzen et al., 2024*) | AI in large-scale screening | Limited sample size | Early diagnosis |
| AI-Assisted Breast Cancer Diagnosis (*Chan, 2024*) | Focus on early detection | High false-positive rate | Better early detection |
| AI-Assisted Risk Assessment (*Thompson, 2023*) | Risk prediction model | No real-time feedback | Personalized screening |
| AI in Mammogram Image Segmentation (*Jha AM & Abraham, 2024*) | Image segmentation | May require large datasets | High-resolution segmentation |

would not only enable fair comparisons between different AI models but also establish a foundation for selecting the most suitable algorithm for specific clinical contexts. Furthermore, the need for interpretable AI outcomes cannot be understated. Radiologists' confidence in AI-generated results relies on the transparency of the decision-making process, thus necessitating the development of methods that provide insights into how AI arrives at its conclusions.

## Diversity in data and generalization

While some studies, such as *McKinney et al. (2020)*, demonstrate the cross-national applicability of AI-assisted mammography, the underlying diversity and representativeness of training data remain critical. AI models are only as effective as the data they are trained on. The challenges of underrepresented populations and variations in healthcare practices between different regions can lead to biases and reduced generalizability. Therefore, addressing these challenges requires a concerted effort in data collection and curation, ensuring that the benefits of AI extend across diverse patient populations. Achieving this goal will contribute to a more equitable distribution of healthcare outcomes.

### Interpretability and trustworthiness

One of the persistent challenges in AI-assisted mammography lies in the interpretability of AI-generated results. Radiologists, as the primary decision-makers, require not only accurate predictions but also an understanding of the reasoning behind those predictions (*Ongena et al., 2021*; *Geras, Mann & Moy, 2019*). To gain their trust and acceptance, AI models must be accompanied by interpretable mechanisms that allow clinicians to comprehend and corroborate the findings. Techniques such as attention maps and saliency maps offer avenues for visualizing the areas of mammograms that AI systems focus on when making decisions. These tools not only provide insights into the features guiding the AI's predictions but also facilitate discussions between radiologists and AI systems, leading to more informed diagnoses.

### Ethical and regulatory considerations

As AI-assisted mammography inches closer to clinical integration, ethical and regulatory considerations loom large. Ensuring patient privacy, obtaining informed consent for AI-generated diagnoses, and addressing potential algorithmic biases are integral components of responsible AI deployment. The dynamic nature of AI algorithms, which can evolve and adapt over time, raises questions about the stability and reliability of these systems. Striking the right balance between fostering innovation and maintaining ethical standards is crucial to avoid unintended consequences and ensure that patients receive accurate and unbiased diagnoses.

### Augmented intelligence and future prospects

The vision of augmented intelligence, where AI complements radiologists' skills, heralds an era of collaborative healthcare decision-making. AI tools can expedite image analysis, identify subtle patterns, and assist in diagnosis, allowing radiologists to focus on complex cases and patient care (*Hickman et al., 2022*). This synergy empowers radiologists with data-driven insights and augments their decision-making abilities. However, achieving this collaboration requires extensive training for radiologists to understand AI-generated results, enabling them to effectively leverage AI's capabilities in their clinical practice. Striking the right balance between human expertise and AI support is essential to ensure optimal patient outcomes.

### Continued research and global impact

While the current studies showcase the potential of AI-assisted mammography, ongoing research is indispensable. Addressing algorithmic biases, evaluating AI's long-term impact on patient outcomes, and validating models on diverse populations are vital steps to ensure that AI tools remain safe and effective (*Hickman et al., 2022*; *Wanders et al., 2022*; *Kohli & Jha, 2018*). Additionally, considering the global impact of AI integration is imperative. Healthcare systems, infrastructure, and resources vary across regions, influencing the feasibility and implications of AI adoption. Tailoring AI approaches to accommodate these differences is essential to ensure that advancements in breast cancer diagnosis are accessible to all.
## Practical clinical implementation of AI-assisted mammography

AI-assisted mammography holds the potential to significantly enhance the accuracy and efficiency of breast cancer detection. Practical clinical implementation involves several critical steps:

- **Data Quality and Diversity**: To integrate AI effectively into clinical practice, robust and diverse datasets are essential. AI models must be trained on mammograms representing various patient demographics (*e.g.*, different breast compositions, ages, and ethnicities) to ensure their generalizability and minimize biases. In practice, this involves collaboration between healthcare institutions to create large, well-annotated datasets that reflect the diversity of clinical conditions.

- **Augmented Intelligence**: AI should be viewed as a tool that augments the radiologist's decision-making process rather than replacing it. This is achieved by implementing AI systems that act as a "second reader" in mammographic interpretation, providing a second layer of validation. Radiologists can focus on complex cases while relying on AI to handle routine analyses, improving overall diagnostic accuracy and reducing false negatives.

- **Interpretability and Trust**: Implementing AI systems requires ensuring that these models are interpretable to radiologists and clinicians. Techniques like saliency maps and feature importance scores can be incorporated into the workflow, allowing clinicians to understand why AI reached certain conclusions. This improves trust in AI-generated diagnoses and enables better decision-making.

- **Validation and Ongoing Monitoring**: Before AI systems are widely adopted, they must undergo rigorous validation, involving clinical trials that test their effectiveness in real-world settings. Post-deployment, continuous monitoring is necessary to ensure that the system maintains its performance and adapts to new challenges in mammography. This may involve regular audits and updates to the model based on evolving clinical data.

- **Integration into Radiology Workflows**: For seamless implementation, AI tools need to be integrated into existing radiology workflows without disrupting clinical operations. This involves providing intuitive user interfaces that radiologists can easily navigate and ensuring that the AI-generated outputs align with radiologists' existing practices.

## Ethical concerns

The ethical implications of AI integration in mammography must be carefully considered to ensure that the technology is used responsibly and equitably.

- **Bias and Fairness**: One of the main ethical concerns is bias in AI algorithms, which can arise if the training data is not representative of the broader population. Addressing this requires building diverse datasets that include a wide range of patient demographics to reduce the risk of biased outcomes and ensure equitable care across all population groups.

- **Data Privacy and Security**: AI models require access to large amounts of patient data, raising concerns about patient privacy and data security. Strict data governance

policies must be in place to anonymize and secure patient information. Techniques like differential privacy and secure multi-party computation can ensure that patient data is protected during model training and inference.

- **Informed Consent**: Patients should be informed when AI systems are being used in their diagnosis. This includes explaining the role of AI in the decision-making process, its benefits, and its limitations. Informed consent processes must be adapted to include AI's role in clinical care, ensuring transparency and maintaining patient trust.
- **Liability and Accountability**: Determining liability in cases where AI-assisted diagnoses result in errors or misdiagnoses is a complex ethical issue. Clear guidelines need to be established regarding the accountability of AI developers, radiologists, and healthcare institutions. A collaborative effort involving legal experts, ethicists, and policymakers is essential to establish appropriate regulatory frameworks.
- **Transparency and Interpretability**: Ethical concerns also extend to the transparency of AI systems. Clinicians and patients alike have a right to understand how AI reached a particular diagnosis. Developing interpretable AI models that radiologists can understand and explain to patients is crucial for fostering trust in the technology.
- **Regulatory Approval**: AI systems must undergo thorough regulatory scrutiny before being deployed in clinical settings. Regulatory agencies, such as the U.S. Food and Drug Administration (FDA) or European Medicines Agency (EMA), are responsible for ensuring that AI models meet strict standards of safety and effectiveness, ensuring that these technologies are used responsibly.

## Future implications

The current landscape of AI-assisted mammography foreshadows a paradigm shift in breast cancer diagnosis, offering profound implications for clinical practice, patient care, and healthcare as a whole. As the capabilities of AI continue to evolve, several significant future implications emerge:

The integration of AI-assisted mammography into routine clinical practice holds immense potential for reshaping radiology workflows. The deployment of AI tools as a second reader or diagnostic aid has the capacity to expedite image analysis and reduce radiologist workload (*Fazal et al., 2018*). This can lead to faster turnaround times, enabling timely diagnoses and treatment decisions. Moreover, the collaborative approach between radiologists and AI systems, where AI acts as a complementary assistant rather than a replacement, aligns with the concept of augmented intelligence. This collaborative framework promises to augment radiologists' capabilities, enhance diagnostic accuracy, and ultimately improve patient outcomes.

AI's ability to analyze complex data patterns opens avenues for tailoring treatment strategies to individual patient characteristics. By leveraging the wealth of data available, AI-assisted mammography can facilitate the identification of subtle trends and correlations that may inform personalized treatment plans (*Fazal et al., 2018*; *Engelman et al., 2010*). Radiologists can benefit from AI-generated insights, allowing them to make informed decisions about treatment options that are best suited to each patient's unique profile.

This shift towards personalized treatment not only enhances patient outcomes but also contributes to the realization of precision medicine in breast cancer care.

While the current studies provide a glimpse into AI's potential, future implications hinge on rigorous research and validation efforts. Longitudinal studies assessing the sustained impact of AI-assisted mammography on patient outcomes are essential to establish its long-term effectiveness (*Sechopoulos & Mann, 2020*). These studies should encompass diverse patient populations, varying clinical contexts, and different stages of breast cancer. Such comprehensive validation is crucial to build robust evidence that supports the incorporation of AI into clinical guidelines and standard practices.

The ethical considerations associated with AI deployment demand heightened attention as AI-assisted mammography progresses towards widespread adoption. The development of transparent and interpretable AI models addresses the need for clinicians to understand AI-generated results and foster trust (*Allen et al., 2021*). Furthermore, ethical frameworks should encompass guidelines for obtaining informed patient consent, ensuring data privacy, and mitigating algorithmic biases. Regulatory bodies and healthcare institutions play a pivotal role in shaping the responsible integration of AI into clinical workflows, safeguarding patient rights, and maintaining the highest standards of patient care.

The future implications of AI-assisted mammography extend beyond technological advancements. The global impact of AI integration requires careful consideration of healthcare disparities, resource availability, and cultural nuances across different regions (*Jerome-D'Emilia, Gachupin & Suplee, 2019*; *Freeman et al., 2021*). While developed healthcare systems might readily embrace AI, developing regions may face challenges in terms of infrastructure, data availability, and AI adoption. Tailoring AI approaches to suit the unique needs of different healthcare contexts ensures that advancements in breast cancer diagnosis are inclusive and equitable.

The landscape of breast cancer diagnosis is undergoing a profound transformation with the integration of AI-assisted mammography. The convergence of advanced machine learning techniques and medical imaging is poised to reshape clinical practice, enhance patient care, and advance our understanding of this complex disease (*Larsen et al., 2022*; *Fusco et al., 2021*). This narrative review has illuminated the key facets of AI's potential, challenges, and future implications in breast cancer diagnosis. The amalgamation of AI algorithms with mammographic images has demonstrated the potential to significantly improve diagnostic accuracy. Studies employing deep learning models have exhibited commendable sensitivity and specificity levels, outperforming traditional methods in detecting breast cancer at early stages. This shift has the potential to redefine the diagnostic journey, leading to more timely interventions and ultimately improving patient outcomes. While the promise of AI is evident, the challenges are equally noteworthy. The dependency on high-quality training data, concerns about algorithmic bias, and the need for interpretability underscore the complexity of AI-assisted mammography (*Ndikum-Moffor et al., 2013*; *Le et al., 2019*). Addressing these challenges requires multidisciplinary collaboration, incorporating inputs from radiologists, data scientists, ethicists, and regulatory bodies. A comprehensive approach ensures that AI's benefits are maximized while minimizing potential pitfalls. The future of breast cancer diagnosis is inherently

collaborative, wherein AI serves as a valued partner to radiologists rather than a replacement. The concept of augmented intelligence emerges as a guiding principle, empowering radiologists with AI-generated insights while upholding their clinical expertise. This synergy fosters an environment where patients receive the highest level of care through the combined strengths of human skill and technological precision (*Le Boulc'h et al., 2020*; *Shen et al., 2019*; *Watson-Johnson et al., 2011*). As AI-assisted mammography inches closer to clinical integration, ethical considerations stand at the forefront. Striking a balance between innovation and patient privacy, ensuring informed consent, and addressing algorithmic biases are vital for responsible AI deployment. The patient-centered approach remains paramount, ensuring that AI tools enhance individualized treatment plans and contribute to improved patient outcomes. The journey of AI-assisted mammography is one of ongoing exploration. Continued research, validation studies, and long-term outcome assessments are pivotal in establishing the sustained efficacy of AI in clinical practice. Moreover, the global impact of AI adoption necessitates sensitivity to regional disparities, ensuring that advancements are accessible to diverse healthcare settings.

### Data diversity and quality

To enhance the accuracy and fairness of AI models in mammography, it is essential to foster collaborative data collection by encouraging partnerships between healthcare institutions globally. This approach will help build comprehensive datasets that encompass diverse demographics, breast compositions, and pathological variations, ultimately reducing bias in AI systems (*Dlamini et al., 2020*). Additionally, implementing data augmentation techniques, such as simulating different imaging scenarios, will artificially expand dataset diversity, ensuring that AI models can generalize across a broader range of clinical conditions. Furthermore, collaboration across institutions to standardize image acquisition protocols is crucial for ensuring consistency in data collection, which will reduce noise in AI training data and improve model performance.

### Generalization of AI models

To ensure AI models in mammography perform effectively in real-world scenarios, it is crucial to incorporate multi-institutional data by using datasets from diverse geographic locations, imaging devices, and clinical settings (*Ginsburg et al., 2020*). This approach enhances the robustness of algorithms beyond controlled environments. Additionally, continuous monitoring post-deployment is essential, with feedback loops enabling radiologists to provide real-time input on AI performance in clinical practice. This ongoing feedback allows AI systems to learn and adapt to new conditions, ensuring they remain accurate and relevant over time.

### Interpretability

To facilitate the integration of AI into clinical workflows, it is essential to develop explainable AI methods, such as attention maps, saliency maps, and feature importance scores, which help radiologists understand why AI algorithms make specific predictions. These tools will enhance transparency and increase confidence in AI-driven decisions (*Batchu et al., 2021*). Additionally, simplifying model architectures can strike a balance between complexity

and interpretability, making AI systems more transparent to radiologists and avoiding the "black-box" problem, where the inner workings of the model are not easily understood by end users.

### Funding
The open access funding for this research is provided by Qatar National Library. The funders had no role in study design, data collection and analysis, decision to publish, or preparation of the manuscript.

### Grant Disclosures
The following grant information was disclosed by the authors:
Qatar National Library.

### Competing Interests
The authors declare there are no competing interests.

### Author Contributions
- Daksh Dave conceived and designed the experiments, prepared figures and/or tables, and approved the final draft.
- Adnan Akhunzada performed the experiments, authored or reviewed drafts of the article, and approved the final draft.
- Nikola Ivković analyzed the data, performed the computation work, prepared figures and/or tables, and approved the final draft.
- Sujan Gyawali analyzed the data, performed the computation work, prepared figures and/or tables, and approved the final draft.
- Korhan Cengiz performed the computation work, prepared figures and/or tables, and approved the final draft.
- Adeel Ahmed conceived and designed the experiments, authored or reviewed drafts of the article, and approved the final draft.
- Ahmad Sami Al-Shamayleh performed the experiments, performed the computation work, prepared figures and/or tables, authored or reviewed drafts of the article, and approved the final draft.

### Data Availability
This is a literature review and did not utilize raw data.

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
