# Peer review of "Diagnostic test accuracy of AI-assisted mammography for breast imaging: a narrative review"

_PeerJ Computer Science, doi:10.7717/peerj-cs.2476_

## Round 0.1 · original submission · Minor Revisions

Dear authors,

Thank you for submitting your Literature Review article. Feedback from the reviewers is now available. It is not recommended that your article be published in its current format. However, we strongly recommend that you address the issues raised by the reviewers and resubmit your paper after making the necessary changes. Before submitting the paper following should also be addressed:

1. Please provide a clearly defined research question for this literature review paper; Differences from the review articles published within this topic should also be provided.
2. Clearly reported, reproducible, and systematic methods should be provided in order to identify, select, and critically appraise relevant research.
3. The Abstract should be attractive and contain motivation.
4. The Introduction section should adequately introduce the subject and make it clear who the audience is and what the motivation is.

Best wishes,

**Language Note:** The review process has identified that the English language must be improved. PeerJ can provide language editing services - please contact us at [email protected] for pricing (be sure to provide your manuscript number and title). Alternatively, you should make your own arrangements to improve the language quality and provide details in your response letter. – PeerJ Staff

Reviewer 1 ·

Basic reporting

The manuscript is centered on a very interesting and timely topic, which is also quite relevant to the themes of the Journal.
Organization of the paper is good and the proposed method is quite novel.

Experimental design

The length of the manuscript seems about right.

Validity of the findings

The manuscript is centered on a very interesting and timely topic, which is also quite relevant to the themes of the Journal.

Additional comments

The paper, however, does not link well with recent literature on sentiment analysis
appeared in relevant top-tier journals,
The manuscript presents some bad English constructions, grammar mistakes, and
misuse of articles: a professional language editing service

·

Basic reporting

1. Basic Reporting
Clarity and Language: The manuscript is written in clear, professional language. The abstract and introduction clearly describe the importance of AI in mammography and its potential benefits and challenges. However, the document can benefit from some minor language polishing to ensure smooth flow, especially in longer sentences. I would suggest simplifying complex statements in the results and discussion sections for better readability.
Introduction & Background: The introduction is well-structured, providing a solid background and rationale for the study. The authors effectively highlight the growing significance of AI in mammography and provide context with relevant literature.
References: The literature is well-cited and current, covering key studies in AI-assisted mammography. However, some references seem a bit outdated (e.g., older than five years), and it may be beneficial to include the most recent advancements in AI and machine learning in healthcare.
Structure: The structure follows the journal's requirements. Figures and tables are presented clearly, but Figure 1 could benefit from more detailed captions to explain its relevance to the findings.
2. Study Design
Relevance to Scope: The article fits within the scope of PeerJ Computer Science as it examines the intersection of AI and healthcare. The narrative review approach is suitable given the nature of the research question.
Methodology: The methodology of selecting and analyzing relevant articles is adequately described. However, the section on the inclusion/exclusion criteria could be more detailed, particularly regarding the thresholds for diagnostic accuracy metrics and how qualitative analysis was conducted.
Survey Methodology: It is well-organized but lacks specific discussion on how potential biases in the literature were handled. For example, how did the authors account for the variability in AI algorithms and clinical settings across the studies they reviewed?
3. Validity of the Findings
Impact and Novelty: The manuscript successfully addresses the growing application of AI in diagnostic mammography, a crucial area in medical imaging. The review identifies both the potential benefits and current challenges (such as interpretability and generalization of AI models), providing value to the field.
Support for Claims: The conclusions are well-supported by the findings of the reviewed studies. The discussion of challenges like data diversity, algorithmic bias, and interpretability is both comprehensive and well-explained. However, the authors could enhance the discussion on future research directions, especially in terms of practical clinical implementation and addressing ethical concerns.
Conclusions: The conclusions are logical and tied to the original research question. However, the "Future Implications" section could be expanded to explore potential strategies for mitigating identified challenges.
4. General Comments
Strengths: The review effectively synthesizes a large body of research on AI-assisted mammography, offering insights into the performance of various AI techniques. The focus on diagnostic accuracy metrics (sensitivity, specificity) is particularly valuable for readers interested in the technical aspects of AI.
Suggestions for Improvement:
Expand on the data diversity section by discussing more specific cases where AI underperformed due to skewed training datasets. Including more geographical diversity in the studies reviewed would also add depth.
Provide more discussion on interpretability in clinical practice, perhaps with examples of explainable AI techniques.
Consider adding more visual aids (e.g., diagrams that show AI model workflows or decision-making processes) to complement the narrative.

Experimental design

As above

Validity of the findings

As above

---

## Round 0.2 · accepted · Accept

Dear authors,

Thank you for the revision. One of the original reviewers did not respond to the invitation to review the revised paper. Another reviewer thinks that the paper is acceptable for publication. I also think that the paper is sufficiently improved and seems acceptable for publication.

Best wishes,

Reviewer 1 ·

Basic reporting

The work presented by the authors is generally written in clear, concise and professional English, although there are some sections where the language could be further clarified and complex sentences simplified. The literature review is comprehensive and adequate references are provided in the context of the study, but the inclusion of more recent studies could enrich the literature. The structure of the paper, figures and tables are presented in a professional manner, with good resolution of the images and clear enough descriptions. Improved with revision

Experimental design

The study design was adequate and an appropriate methodology was used to answer the stated research questions. The design was structured to ensure that the study achieved its objectives and the necessary steps were clearly explained.Improved with revision

Validity of the findings

The validity of the research findings is positive. The methods and analyses used were applied correctly and the results are consistent. The findings support the objectives of the study and there is consistency between the data set used and the analysis methods.

Additional comments

With the revision, the paper has become publishable